# A Cephalometric Analysis Assessing the Validity of Camper’s Plane to Establishing the Occlusal Plane in Edentulous Patients

**DOI:** 10.3390/dj11030081

**Published:** 2023-03-15

**Authors:** Lina Sharab, David Jensen, Gregory Hawk, Ahmad Kutkut

**Affiliations:** 1Division of Orthodontics, College of Dentistry, University of Kentucky, Lexington, KY 40536, USA; 2Department of Statistics, University of Kentucky, Lexington, KY 40536, USA; 3Division of Prosthodontics, College of Dentistry, University of Kentucky, Lexington, KY 40536, USA

**Keywords:** Camper’s plane, occlusal plane, edentulous, cephalometric X-ray

## Abstract

Complete denture fabrication requires multiple clinical and laboratory steps. One of the most critical clinical steps is establishing an anatomical occlusal plane based on hard and soft tissue references. The aim of this study was to determine whether age or gender affects the level of the Ala-Tragus plane to establish which reference point on the Tragus should be used when fabricating the occlusal plane in edentulous patients. Clinical photographs and lateral cephalometric radiographs with complete dentitions were taken from 58 volunteers at the DMD clinic at the University of Kentucky. Each photograph was superimposed over its corresponding cephalometric image. An analysis was conducted to establish the angle of the occlusal plane relative to the Ala-Tragus landmarks; this data was then grouped according to age and gender. The analysis shows that age and gender did not significantly affect where the Camper’s plane should be approximated for complete denture treatment. However, it was found that the most parallel line to the occlusal plane was Ala’s inferior border to the ‘Tragus’s inferior border. It should be noted that the volunteers’ skeletal classification was significantly related to a Cl III malocclusion tendency. Still, with this new information, functionality and esthetics can be more adequately addressed for patients undergoing complete denture treatment. Given our results, we suggest redefining the ‘Camper’s plane with a line extending from ‘Ala’s inferior border to the ‘Tragus’s inferior border instead of the superior border. Further consideration should be taken if the patient is a skeletal CL III malocclusion.

## 1. Introduction

Recording the maxillomandibular relationship of a jaw to establish an occlusal plane is one of the most critical steps in fabricating a complete denture [1,2]. The occlusal plane is the common plane established by the incisal and occlusal surfaces of the teeth [3]. The orientation of the occlusal plane for edentulous patients is important in achieving denture treatment esthetic [3]. It is crucial for a functionally balanced occlusion for complete denture wearers [3]. According to Boucher, “it seems to be obvious that if the soft tissue surrounding the denture is to work around as they did around natural teeth, the occlusal plane should be oriented as close as it was when the natural teeth were present” [4,5,6]. In other words, the occlusal plane position in denture wearers should simulate what was previously established by a patient’s natural teeth [4].

The occlusal plane forms a basis for the ideal tooth arrangement, fulfilling the necessary esthetic and mechanical requirements and aiding in proper oral function [7,8]. In complete denture fabrication, the location of the occlusal plane can be determined depending on various soft and hard tissue landmarks and the clinician’s clinical judgment [9]. The properly positioned occlusal plane helps the normal function of the tongue and cheek muscles, enhancing denture stability during function [4,8].

The question of the occlusal plane is particularly important to patients. The inclination of an occlusal plane can affect a smile’s esthetic. For instance, orthodontic treatment for cases with dentofacial skeletal discrepancies typically includes changes in the occlusal plane for esthetic improvement, i.e., the Functional Aesthetic Occlusal Plane (FAOP) [10]. In addition to occlusal relation, occlusal plane tipping is related to skeletal facial relations, vertical facial type and anteroposterior skeletal relation. Providing complete denture treatment for growing patients is challenging due to the continuous changes in the skeletal and facial anatomy. Multiple prostheses are needed to maintain the support for facial tissue and enhance the esthetics to accommodate these changes [11,12,13].

Literature provides evidence for several landmarks for clinicians to determine the occlusal plane’s most acceptable position [14,15,16,17,18,19,20,21,22,23,24,25,26,27,28,29]. It has been reported that relocating the occlusal plane may be achieved by different methods, such as by using anterior esthetic considerations and parallel to the Ala-Tragus line posteriorly; positioning it parallel to the residual ridges; orienting it to the buccinator grooves and the commissure of the lips; terminating it posteriorly at the middle or upper third of the retromolar pad, and positioning it on the same level as the lateral border of the tongue. However, none of the previous methods were accurately used in relocating the occlusal plane or proved the consistent accuracy of any particular orientation method [3,4,5,6,7,8,9,10,11,12,13,14,15,16,17,18,19,20,21,22,23,24,25,26,27].

The most common clinical method for orienting the occlusal plane positions is parallel to the Camper’s plane, which is extended from Ala’s inferior border to the Tragus’s superior border [30,31,32,33,34,35,36]. Camper’s plane and Frankfort’s horizontal (FH) plane have been clinically used to establish the occlusal plane for denture fabrication [31]. Unfortunately, this method often creates confusion in selecting a point for marking the Ala-Tragus line. Numerous authors and researchers have proved that all three parts of the Tragus, namely the superior, middle and inferior, can guide occlusal plane orientation in edentulous patients [32,33,34,35,36]. This study aimed to determine whether age or gender affects the level of the Ala-Tragus plane to establish the most appropriate reference point on the Tragus to be used when making the occlusal plane for edentulous patients. Using the Ala-Tragus method, we tested the hypothesis that the occlusal plane has unique associations with different points of the Tragus based on the patient’s age. This validation was performed by examining and collecting data from individuals with complete, unaltered functional dentitions. We used composite soft tissue and cephalometric overlay to establish age/gender norms according to the occlusal plane and the Ala-Tragus landmarks. This data can aid practitioners in establishing the best location of the occlusal table for denture esthetics and function. The specific question for the study was: Dose the Ala-Tragus plane differ based on age and gender?

## 2. Materials and Methods

This study was a single-site, clinical, and radiographic study. A total of 58 volunteers (26 males and 32 females) were recruited from the University of Kentucky College of Dentistry and distributed into two groups based on age: Group A or the Young adult age group, included 30 volunteers between the ages of 22 and 50; Group B was our Old age group and included 28 volunteers over the age of 51. Fifty-eight subjects is a reasonable number for a pilot study. The sample size, along with the observed statistical results and within-subject correlations that we obtained, will prove essential for designing the larger, confirmatory study that constitutes the next phase of this research.

The University of Kentucky, Institutional Review Board, approved the study (IRB Number: 46061). Volunteers at the University of Kentucky, College of Dentistry, who met the study criteria, were invited to participate in this study by the study investigators. The volunteers who agreed to participate in this clinical study, signed the Informed Consent and combined HIPPA forms. In addition, all research team members received training in human subject protection, a training course required by the University of Kentucky. Informed consent was required for this clinical and radiographic research and copies are available as needed.

For each volunteer, 3 extraoral photos were taken from the right, left, and frontal views using a professional camera and one lateral Ceph. An X-ray was made at rest using an Instrumentarium Dental ORTHOCEPH™ OC200 D. The right side of each cephalogram was traced using Dolphin Imaging & Management Solutions software, then superimposed on the right profile photo by one examiner (Figure 1).

We used a FOX plane plate to evaluate the occlusal plane of dentate subjects from all groups. Three points were evaluated on the Tragus as Superior (S), Middle (M), and Inferior (I). These points were joined with the nose’s Ala (A) inferior border to determine the Ala-Tragus lines digitally with the Dolphin software (Figure 2).

The angle formed by each resulting line, referred to as the SA plane, MA plane, and IA plane, was measured from the Ceph X-ray analysis along with the Frankfort Horizontal (FH) plane. All angles between the Frankfort Horizontal and natural occlusal planes were measured by superimposing the Ceph analysis onto clinical photos using Dolphin software. Two landmarks were used to superimpose the Ceph with the photo for proper orientation and sizing: the external auditory meatus and the central incisal edge. The final images were analyzed photometrically. The most parallel relationship was determined between the arms of the FOX plane, which represented the clinical occlusal plane and the three different levels of the Ala-Tragus line.

The volunteers included in this study were 22 years or older and in good general health. They were dentate. Excluded from this study were volunteers who had a previous history of orthodontic or orthognathic treatment [10], no posterior teeth present to determine the occlusal plane, a history of facial or temporomandibular joint surgeries, supraeruption or the drifting of teeth, and volunteers who were pregnant, breastfeeding, or did not read English.

All research activities were conducted at the University of Kentucky College of Dentistry facilities. This study had the usual potential risks associated with study participation, including a potential breach of confidentiality and radiation risks associated with the X-ray. However, the photographs and radiographs were always conducted using Universal Precautions. The knowledge gained from this study could aid in a more accurate orientation of the occlusal plane for optimal esthetics in the prosthodontic rehabilitation of edentulous patients. The risk associated with unnecessary X-ray radiation for Ceph analysis was considered reasonable for research purposes to validate the clinical orientation of the FOX plane.

## 3. Results

Descriptive statistics were used to qualitatively calculate and summarize each variable’s outcome. As was appropriate, the differences between the groups were analyzed using a Welch two-sample *t*-test or one-way ANOVA model. Corresponding pairwise comparisons were calculated for the ANOVA models with a significant overall effect. Multiple comparisons were adjusted using Fisher’s Least Significant Differences (LSD) method. Linear regression models were fit to assess the relationship between the quantitative explanatory variables and the outcome variables. When leveraging the analysis and residual plots suggested, it was necessary for the outlying data points to be removed, and the models re-fitted. Across all analyses, a *p*-value of less than 0.05 was considered significant. All analyses were completed in R, version 4.0.4 (R Foundation for Statistical Computing).

When comparing the difference between the FOX plane and FH across the three levels of the Tragus (S, M, and I), we found each angle to be significantly different from the other [*p* < 0.0001]. The angles formed between ’Ala’s inferior border to Tragus’s inferior border in reference to the FOX plane and FH were significantly smaller than both the middle and superior angles. Furthermore, the angles formed in the middle of the Tragus in reference to FH-FOX were significantly smaller than those formed from the superior border of the Tragus in reference to FH-FOX. So the most parallel Ala-Tragus plane to the FOX plane is the inferior border of the Tragus (Table 1 and Figure 3).

For statistical reasons, the volunteers were assigned to two skeletal class tendencies, 50% CL II or CL I with CLL II tendency and 50% CL III. Among the subjects, the angle that measured the relative position of the maxilla to the mandible, to determine the anteroposterior skeletal relationship ANB to the distribution of skeletal tendencies, was fairly normal, ranging from −3 to 7 degrees (Figure 4).

Subjects were also grouped into two categories of vertical tendencies: one group had a 33% tendency to a high Mandibular Plane Angle (MPA), which describes the degree of steepness of the mandible to the face; the second had a 67% tendency to a low MPA (Figure 5). It was found that the relationship between the FOX plane and Ala-Tragus inferior and middle reference points had significantly more counterclockwise tipping for individuals with CL III tendency compared to CL II tendency ([*p* = 0.009] [*p* = 0.048], respectively) (Table 2). This did not hold for the superior location, where [*p* = 0.334] (Table 3). In addition, FOX measurements had significantly more forward tip (clockwise) for individuals with increased facial vertical tendencies compared to low vertical tendencies for all three locations [superior *p* = 0.022, inferior *p* = 0.0002, middle *p* = 0.001]. This indicates steeper downward tipping of the natural occlusal plane (OP) in individuals with a high mandibular plane angle (Table 4).

The difference between the FOX–AT relationships measured from photos and the OP-FH relationships measured from the Ceph were essentially the same. Therefore, this analysis verified the match between the occlusal plane on the Ceph and the clinical occlusal plane with the FOX plane on the photos, with a mean difference of 1.4 degrees. In regard to the upper incisor angulation to FH (flaring), as the teeth were more flared, the FOX planes were more likely to tip upward anteriorly [superior *p* = 0.043, inferior *p* = 0.0006, middle *p* = 0.031]. This makes sense from a geometric standpoint and may be a concern for CL III patients. The degree of tipping of the OP may influence the esthetic outcome of the patient’s smile with the dentures.

## 4. Discussion

The natural position of an occlusal plane is crucial for esthetics, phonetics and lost vertical dimension that can be established by the teeth incisal edges and occlusal surfaces [34,35,36,37]. For edentulous patients, placing denture teeth in the natural position enables oral muscles and other surrounding structures to function normally [38,39]. Using radiographic and 3D methods with various levels of invasiveness, this study determined an effective and consistent method to obtain accuracy in reconstructing the occlusal plane [40,41]. Previous studies, in contrast, showed differences in angulations of the occlusal plane with Camper’s plane. For example, previous studies found the angulations of the occlusal plane to Camper’s plane as 3.45°, 7.00° and 10.00°, respectively [18,42,43]. On the other hand, the occlusal plane-FH plane angulation was reported as 11.42° in dentulous subjects [37]. In contrast, Celebic et al. proposed 9.43° and 8.53° in dentulous and edentulous subjects, respectively [44].

Prosthodontists have been controversial regarding using the Camper’s plane or the Ala-Tragus line to establish the occlusal plane in edentulous patients. Which reference point of the Tragus has to be considered? Is it the Tragus’s superior border or the Tragus’s middle or Tragus’s inferior border?

To solve this controversy, the present investigation was undertaken. First, we performed cephalometric tracing on dentulous patients with different age groups and looked into malocclusions to apply our results to edentulous patients. We used adult volunteer subjects since the orientation of the occlusal plane fluctuates with growth [8,43,45,46,47]. Cephalometric tracings were performed on lateral cephalograms obtained from these volunteers. The parallelism between the Camper’s line and the occlusal plane was checked on the three points: the superior, middle and inferior borders of the Tragus of the ear.

The results have shown no statistically significant difference between the Young and Old age groups regarding the three Ala-Tragus reference points, namely the superior, middle, or inferior points. However, the smallest angle discrepancy with the occlusal plane was shown with the inferior border of the Tragus (6.2° ± 7.4°). Our results agree with the work of Singh et al. [48], which also revealed that the line drawn from the inferior border was much more parallel to the occlusal plane of the dentulous patient than the middle and superior borders [47]. Our results also confirm the findings of previous work, which relates the orientation of the natural occlusal plane to the vertical skeletofacial patterns for both vertical and anteroposterior skeletal angles [20]. However, clinicians must consider evaluating the esthetics based on the Camper’s plane and the various anatomical landmarks of each patient. There was no statistically significant difference between genders regarding the Tragus’ superior, middle, or inferior borders. Most studies in the literature showed that the FH plane is a reliable skeletal landmark; however, Camper’s plane is also a reliable clinical landmark for establishing the lost occlusal plane in edentulous subjects with various jaw relationships [34,35]. This study shows that the Camper’s, FH and PL planes are practical guides for establishing an occlusal plane. Therefore, using the Camper’s plane as a landmark for establishing the occlusal plane along with FB to transfer the FH plane can offer accurate results for the occlusal plane’s establishment. It should be noted that even with the best measures, there is no approved scientific method for establishing the occlusal plane. Therefore, it is a highly subjective relation.

The digital denture is an emerging technology that has modified the conventional way of fabricating complete dentures. Three-dimensionally printed, or CAD/CAM milled dentures, are a promising technology that has accelerated the treatment process by reducing the number of visits and allowing for higher quality materials and esthetics [49]. Digital articulation, using artificial intelligence, has eliminated the need for facebow transfer and physical mounting. However, conventional and digital techniques still require the establishment of the anatomical occlusal plane for complete denture patients [50,51].

### Clinical Implications

Cephalometric analysis of dentate subjects was used to assess the Ala-Tragus line’s validity in different age groups and genders. The results of this investigation may assist clinicians in determining appropriate landmarks on the Ala-Tragus’s inferior border instead of the Tragus’s superior border to establish the appropriate occlusal plane for edentulous patients, i.e., Ala’s inferior border to the Tragus’s inferior border. In addition, with this improved method of denture reconstruction, clinicians will be better able to improve the functionality of denture work and esthetics.

## 5. Conclusions

Given our results, we suggest redefining the Camper’s plane with a line extending from Ala’s inferior border to the Tragus’s inferior border instead of the superior border. Further consideration should be taken if the patient is a skeletal CL III malocclusion.

## Figures and Tables

**Figure 1 dentistry-11-00081-f001:**
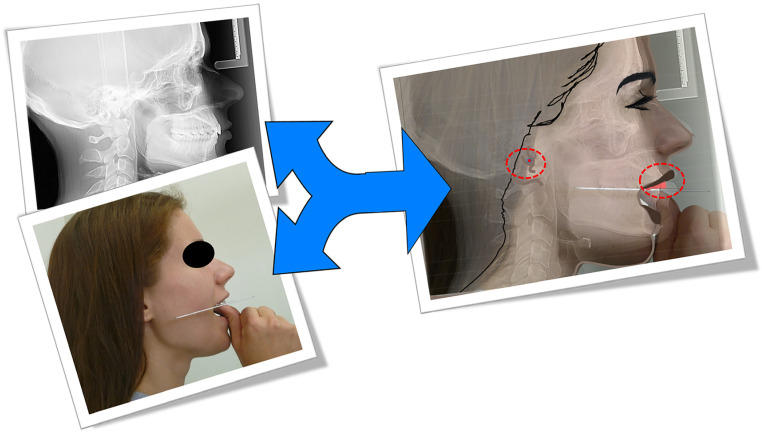
The right side of each cephalogram was traced using the Dolphin program and superimposed on the right profile photo. The red circles indicate the reference points for the occlusal plane.

**Figure 2 dentistry-11-00081-f002:**
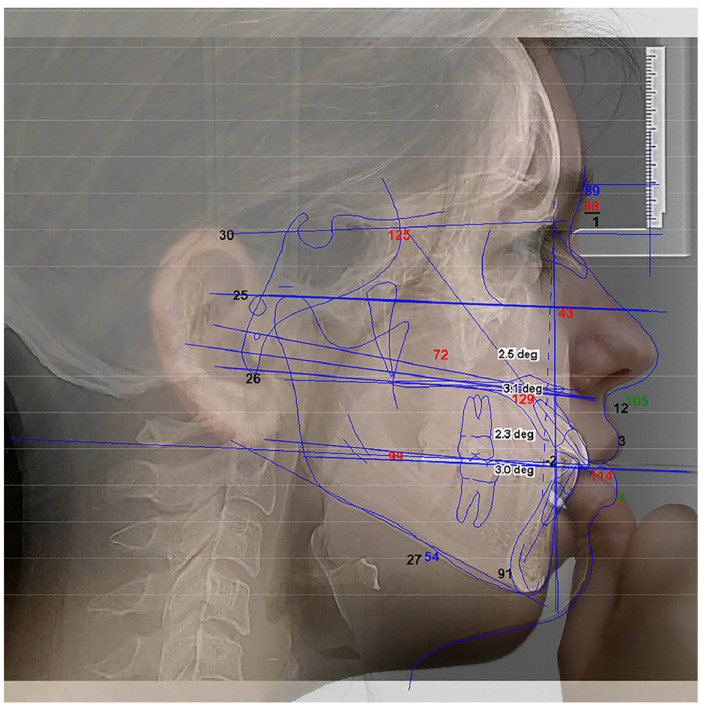
The FOX plane was used clinically to determine the occlusal plane of dentate subjects. Three points were evaluated on the Tragus as Superior (S), Middle (M), and Inferior (I). The points were joined with the inferior border of the Ala (A) of the nose to form Ala-Tragus lines digitally using Dolphin software(© 1998-2020 Dolphin Imaging and Management Solutions, CA, USA, version 11.95 SP).

**Figure 3 dentistry-11-00081-f003:**
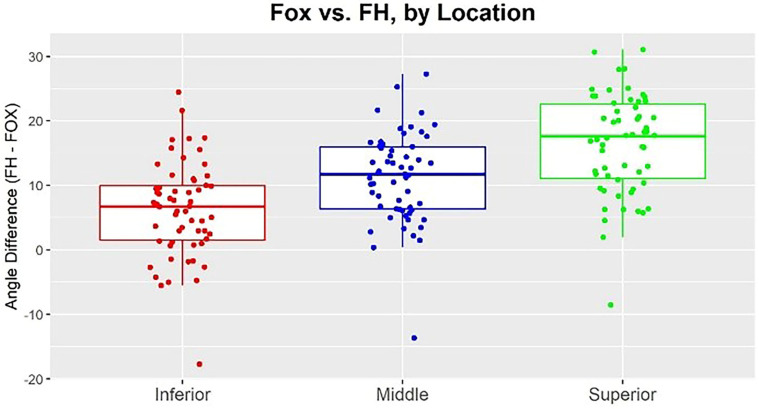
The most parallel Ala-Tragus planes to the FOX plane is with the inferior border of the Tragus.

**Figure 4 dentistry-11-00081-f004:**
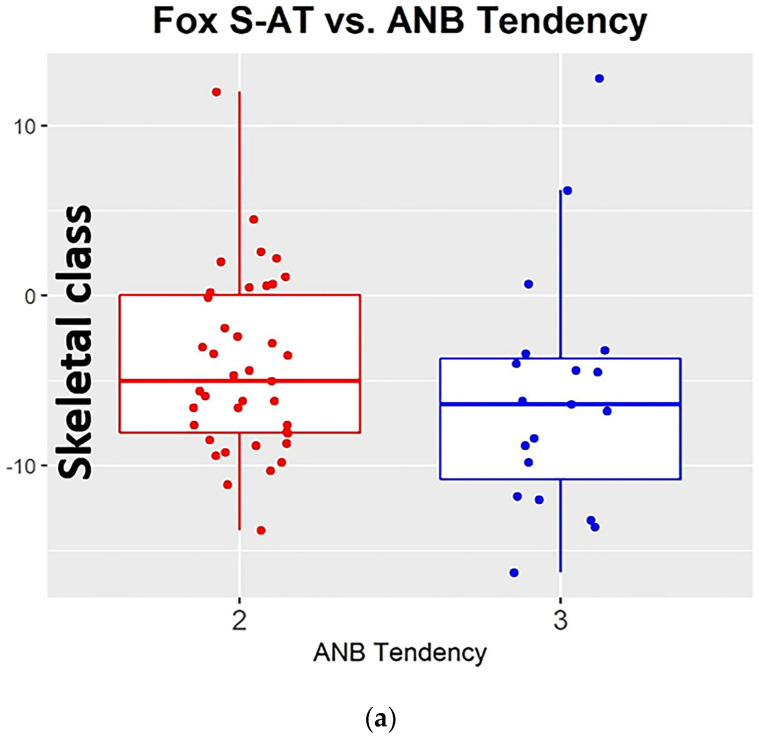
The distribution of skeletal tendencies for the ANB angle: (**a**) for the Tragus superior border, (**b**) for the middle point, and (**c**) for the inferior border.

**Figure 5 dentistry-11-00081-f005:**
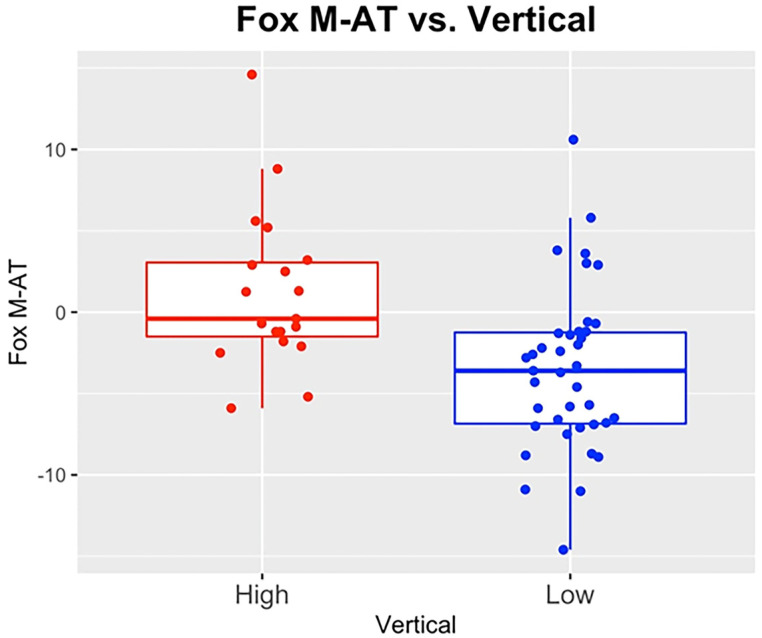
The tendency to high or low MPA.

**Table 1 dentistry-11-00081-t001:** Comparing the difference between the FOX plane and FH across the 3 levels of the Tragus (S, M, I), the most parallel Ala-Tragus plane to the FOX plane is with the inferior border of the Tragus as the smallest angle.

Location	Mean ± SD
Superior	16.4° ± 7.8°
Middle	11.2° ± 6.9°
Inferior	6.2° ± 7.4°

**Table 2 dentistry-11-00081-t002:** The estimated mean FOX I-AT for ANB tendency Cl III is 3.82 degrees more negative than the estimated mean FOX I-AT for ANB tendency II [*p*-value = 0.009].

ANB Tendency	Mean ± SD
II (*n* = 39)	1.7 ± 5.1
III (*n* = 19)	−2.1 ± 4.9

**Table 3 dentistry-11-00081-t003:** The estimated mean FOX M-AT for ANB tendency III is 3.10 degrees more negative than the estimated mean FOX M-AT for ANB tendency II [*p*-value = 0.048].

ANB Tendency	Mean ± SD
II (*n* = 39)	−1.0 ± 5.1
III (*n* = 19)	−4.1 ± 5.6

**Table 4 dentistry-11-00081-t004:** The estimated mean FOX M-AT for low verticality is 4.79 degrees more negative than the estimated mean FOX M-AT for high verticality [*p*-value = 0.001].

Vertical	Mean ± SD
High (*n* = 19)	1.2 ± 4.9
Low (*n* = 39)	−3.6 ± 5.0

## Data Availability

Informed consent was required for this clinical and radiographic research and is available as needed.

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
