# Peer review of "A Cephalometric Analysis Assessing the Validity of Camper’s Plane to Establishing the Occlusal Plane in Edentulous Patients"

_dentistry, 2023, doi:10.3390/dj11030081_

Round 1
Reviewer 1 Report
Quite interesting work on the correspondence between occlusal planes and cephalometric analyses.
Sufficient quality work with some criticisms below:
-The abstract section should begin with a general statement on the issue that led to the study
-line 17: remove statement relating to the ethics committee
-Check that all keywords are pubmed mesh terms
-Line 55 in this section also insert the importance of determining the occlusal planes in growing patients in need of orthodontic therapies. In this regard, I suggest to insert in the reference section the following scientific work that could be of help to the reader:
Giuca MR, Pasini M, Drago S, et al. Influence of Vertical Facial Growth Pattern on Herbst Appliance Effects in Prepubertal Patients: A Retrospective Controlled Study. Int J Dent. 2020;2020:1018793. Published 2020 Jan 11. doi:10.1155/2020/1018793
- Define in detail the training and calibration procedures of the medical staff for the execution of the study
- a section dedicated to the statistical analysis used in the study is missing
- some considerations are missing in the discussion section on the role that 3D technology can bring in terms of diagnosis and dental therapy, both in the prosthetic, gnathological and orthodontic fields. In this regard, I suggest to insert in the reference section the following scientific work that could be of help to the reader:
Valenti C, Isabella Federici M, Masciotti F, et al. Mechanical properties of 3D-printed prosthetic materials compared with milled and conventional processing: A systematic review and meta-analysis of in vitro studies [published online ahead of print, 2022 Aug 5]. J Prosthet Dent. 2022;S0022-3913(22)00415-2. doi:10.1016/j.prosdent.2022.06.008
-A comprehensive review of the English language is required from the authors
Author Response
Reviewer 1:
Quite interesting work on the correspondence between occlusal planes and cephalometric analyses.
AK: Thank you for the feedback.
Sufficient quality work with some criticisms below:
-The abstract section should begin with a general statement on the issue that led to the study
AK: The following statement was added to the abstract. Complete denture fabrication requires multiple clinical and laboratory steps. One of the most critical clinical steps is establishing an anatomical occlusal plane based on hard and soft tissue references.
-line 17: remove statement relating to the ethics committee
AK: The statement relating to the ethics committee was removed.
-Check that all keywords are pubmed mesh terms
AK: Keywords are confirmed with pubmed mesh terms and modified
-Line 55 in this section also insert the importance of determining the occlusal planes in growing patients in need of orthodontic therapies. In this regard, I suggest to insert in the reference section the following scientific work that could be of help to the reader:
Giuca MR, Pasini M, Drago S, et al. Influence of Vertical Facial Growth Pattern on Herbst Appliance Effects in Prepubertal Patients: A Retrospective Controlled Study. Int J Dent. 2020;2020:1018793. Published 2020 Jan 11. doi:10.1155/2020/1018793
AK: The outdated reference # 13 was replaced with the suggested reference, and the text was modified. The following statement was added to show the correlation of the importance of occlusal planes in growing patients: Providing complete denture treatment for growing patients is a challenging due to the continuous changes in the skeletal and facial anatomy. Multiple prostheses are needed to maintain the support for facila tissue and enhance the aesthetics to accommodate these changes.
- Define in detail the training and calibration procedures of the medical staff for the execution of the study
AK: all research team members received training in human subject protection, a training course required by the University of Kentucky.
- a section dedicated to the statistical analysis used in the study is missing
AK: The statistician “Mr. Gregory S. Hawk” was involved in the study design and completed all statistical analysis with the result interpretations.
- some considerations are missing in the discussion section on the role that 3D technology can bring in terms of diagnosis and dental therapy, both in the prosthetic, gnathological and orthodontic fields. In this regard, I suggest to insert in the reference section the following scientific work that could be of help to the reader:
Valenti C, Isabella Federici M, Masciotti F, et al. Mechanical properties of 3D-printed prosthetic materials compared with milled and conventional processing: A systematic review and meta-analysis of in vitro studies [published online ahead of print, 2022 Aug 5]. J Prosthet Dent. 2022;S0022-3913(22)00415-2. doi:10.1016/j.prosdent.2022.06.008
AK: The following paragraph was added: Digital denture is an emerging technology that modified the conventional way of fabricating complete dentures. 3D printed, or CAD/CAM milled dentures, is a promising technology that accelerated the treatment process by reducing the number of visits and allowing for higher quality materials and esthetics. [49] The digital articulation using artificial intelligence eliminated the need for facebow transfer and physical mounting. However, conventional and digital techniques still require establishing the anatomical occlusal plane for complete denture patients. [50,51]
-A comprehensive review of the English language is required from the authors
AK: Thank you for the suggestion. The major English editor revised the manuscript.

Reviewer 2 Report
Dear authors,
I have reviewed your study, "A Cephalometric Analysis Assessing the Validity of Camper's 2 Plane to Establishing the Occlusal Plane in Edentulous Patients", and I appreciate the detailed methodology used to investigate the validity of Camper's 2 Plane for establishing the occlusal plane in edentulous patients. The study was well-structured and designed, and the statistical analysis was appropriate and well-explained. However, I noticed a few issues that need to be addressed.
In the abstract, the purpose of the study was not clearly stated, and it would be helpful to clarify the specific research question that the study aims to answer in the abstract. The sample size of 58 volunteers may be considered small, and it would be useful to address whether this sample size was adequate to provide statistically significant results. Additionally, it would be helpful to know if the study investigators were blinded to the participants' age and gender during the analysis, as this information could potentially influence the results. The results of the study are clearly presented and discussed, and the tables and figures are helpful in illustrating the findings. However, the authors should provide more context for the statistical analyses used, as some readers may not be familiar with the Welch two-sample t-test, one-way ANOVA model, Fisher's Least Significant Differences method, and linear regression models. Additionally, the interpretation of statistical significance should be more explicitly stated, as some statements suggest that significance was found when it may not have been.
One major concern I have is the statement in the abstract that "age and gender did not significantly affect where the Camper's plane should be approximated for complete denture treatment." This conclusion may be misleading, as the results indicate that age and gender did not significantly affect the level of the ala-tragus plane, but they do not necessarily suggest that the Camper's plane should not be adjusted based on age and gender. It would be helpful to clarify this point in the discussion section and consider providing recommendations for clinical practice based on the findings. Overall, I found your study to be well-designed and well-executed, with clear results and discussion. I suggest that you address the minor and major issues mentioned above to improve the clarity and applicability of your findings.
Thank you for your hard work in conducting this study, and I look forward to seeing the revisions.
Author Response
Reviewer 2:
Comments and Suggestions for Authors
Dear authors,
I have reviewed your study, "A Cephalometric Analysis Assessing the Validity of Camper's 2 Plane to Establishing the Occlusal Plane in Edentulous Patients", and I appreciate the detailed methodology used to investigate the validity of Camper's 2 Plane for establishing the occlusal plane in edentulous patients. The study was well-structured and designed, and the statistical analysis was appropriate and well-explained. However, I noticed a few issues that need to be addressed.
AK: Thank you for the feedback.
In the abstract, the purpose of the study was not clearly stated, and it would be helpful to clarify the specific research question that the study aims to answer in the abstract.
AK: The specific question for the study was: Dose the Ala-Tragus plane differ based on age and gender? And added to the text.
The sample size of 58 volunteers may be considered small, and it would be useful to address whether this sample size was adequate to provide statistically significant results.
AK: 58 subjects is reasonable for a pilot study. This means, along with the observed standard deviations and within-subject correlations that we obtained, will prove essential for designing the larger, confirmatory study that constitutes the next phase of this research.
Additionally, it would be helpful to know if the study investigators were blinded to the participants' age and gender during the analysis, as this information could potentially influence the results.
AK: The study investigator who analyzed the cephalometric x-rays and clinical photos was not blinded. However, the measurements were analyzed using Dolphin software which has no bias on age or gender.
The results of the study are clearly presented and discussed, and the tables and figures are helpful in illustrating the findings. However, the authors should provide more context for the statistical analyses used, as some readers may not be familiar with the Welch two-sample t-test, one-way ANOVA model, Fisher's Least Significant Differences method, and linear regression models. Additionally, the interpretation of statistical significance should be more explicitly stated, as some statements suggest that significance was found when it may not have been.
AK: The statistician “Mr. Gregory S. Hawk” was involved in the study design and completed all statistical analysis with the result interpretations.
One major concern I have is the statement in the abstract that "age and gender did not significantly affect where the Camper's plane should be approximated for complete denture treatment." This conclusion may be misleading, as the results indicate that age and gender did not significantly affect the level of the ala-tragus plane, but they do not necessarily suggest that the Camper's plane should not be adjusted based on age and gender. It would be helpful to clarify this point in the discussion section and consider providing recommendations for clinical practice based on the findings. Overall, I found your study to be well-designed and well-executed, with clear results and discussion. I suggest that you address the minor and major issues mentioned above to improve the clarity and applicability of your findings.
AK: The following statement was added to the discussion: However, clinicians must consider evaluating the esthetics based on the Camper’s plane and various anatomical landmarks of each patient.

Round 2
Reviewer 1 Report
All comments were added
Author Response
Thank you. The minor English revision was completed.
Reviewer 2 Report
Thank you for your submission. I am pleased to inform you that all of the questions have been resolved, and we highly recommend your work for publication.
Author Response
Thank you for the publication recommendation.